# Ethnic inequalities in patient satisfaction with primary health care in England: Evidence from recent General Practitioner Patient Surveys (GPPS)

**John Paul Magadi[1], Monica Akinyi Magadi[2]***

**1** School of Management, University of Bradford, Bradford, United Kingdom, **2** School of Criminology, Sociology and Policing, University of Hull, Hull, United Kingdom

* m.m.magadi@keele.ac.uk

**Data Availability Statement:** The GPPS datasets are available to the interested public/researchers and free to download under 'UK open data' (https://gp-patient.co.uk/surveysandreports).

## Abstract

### Aims and objectives

This paper aims to improve understanding of factors that contribute to persistent ethnic disparities in patient satisfaction in England. The specific objectives are to (i) examine ethnic differences in patient satisfaction with their primary care in England; and (ii) establish factors that contribute to ethnic differences in patient satisfaction.

### Data and methods

The study is based on secondary analysis of recent General Practitioner Patient Survey (GPPS) datasets of 2019, 2020 and 2021. Descriptive bivariate analysis was used to examine ethnic differences in patient satisfaction across the three years. This was followed with multilevel linear regression, with General Practice (GP) at level-1 and Clinical Commissioning Group (CCG) at level-2 to identify factors contributing to ethnic differences in patient satisfaction.

### Results

The findings show consistent negative correlations between the proportion of patients reporting good (very or fairly good) overall experience and each of the ethnic minority groups. Further examination of the distribution of patient satisfaction by ethnicity, based on combined ethnic minority groups, depicted a clear negative association between ethnic minority group and patient satisfaction at both GP and CCG levels. Multilevel regression analysis identified several service-related factors (especially ease of using GP website and being treated with care and concern) that largely explained the ethnic differences in patient satisfaction. Of all factors relating to patient characteristics considered in the analysis, none was significant after controlling for GP service-related factors.

### Conclusions

Ethnic minority patients in England continue to consistently report lower satisfaction with their primary health care in recent years. This is largely attributable to supply (service

**Funding:** The authors received no specific funding for this work.

**Competing interests:** The authors have declared that no competing interests exist.

related) rather than demand (patient characteristics) factors. These findings have important implications for health care system policy and practice at both GP and CCG levels in England.

# 1: Introduction

## 1.1 Study background and rationale

Reducing ethnic inequalities in health has explicitly been part of UK government agenda since 1997 [1]. The 2010 Equalities Act mandated government bodies in the UK to monitor and address inequalities by factoring age, sex, ethnicity and sexual orientation [2]. The government bodies include the national health service (NHS). This was warranted by vast range of studies that consistently determined that marginalized groups based on their age, sex, ethnicity, and sexual orientation were susceptible to receiving a lower quality of service among the various public sectors [3–5]. In 2014 NHS echoed concerns in the differences in the quality of care delivered to groups who were disadvantaged based on who they were [6]. However, there have been limited substantial changes in inequalities in patient experience in primary health care within England during the period 2011 to 2017 [2]. Patient satisfaction is an important indicator of the quality of care and may help relevant policy makers identify and address limitations with the health care system [7, 8]. Good patient experience and satisfaction usually translates into better health outcomes as patients are likely to be more engaged during consultations, better collaborate with General Practice (GP) staff and more likely to adhere to treatment plans [9].

The persistent lower patient satisfaction of ethnic minorities in comparison to the ethnic majorities in England [4] calls for further research to better understand factors influencing lower patient satisfaction so that they can be addressed more effectively. This is especially critical in the current period given dire resource constraints within the NHS. Ethnic minorities general tendency to be disproportionately dissatisfied with state of their health and the primary care they receive could be attributable to range of factors. These may range from genetic factors or tougher living conditions often associated with poorer health [10] that may impact patient satisfaction, to being more likely to experience stress in general, including while receiving primary health care [11]. Furthermore, primary health care in the UK may be typically not fully equipped to deal with the complex health care needs of ethnic minorities, and may be unable to effectively cater for the health needs of a diverse range of ethnic groups with different backgrounds/cultures. Ali et al. [12] noted that NHS services ignored needs specific to ethnic minorities' sections of the community, resulting in poor outcomes.

## 1.2 Factors associated with low patient satisfaction among ethnic minorities

Factors identified from existing literature to be associated with low patient satisfaction among ethnic minorities include social deprivation, communication, health status, access of health services, and demographic/other characteristics. Social deprivation has consistently been identified as a strong determinant of low patient satisfaction, and a likely contributing factor to low satisfaction among ethnic minority patients [3, 13].

Communication has been identified as a key factor in overall patient satisfaction. An earlier analysis of the English GP survey data 2012–2014 by Burt et al. [6] found that groups specifically of older, Pakistani and Bangladeshi female and younger individuals who identified as

other white (other European countries) were associated with issuing the lowest ratings with regards to their doctors' communication. Southeast Asians and younger patients have consistently been linked to reporting lower patient satisfaction scores in other studies [4, 8]. The importance of communication has also been established in studies outside the UK. In a study of barriers to adequate patient participation in consultations with their doctors in the Netherlands, Schinkel et al. [14] observed that the Dutch (ethnic majority) had little to no barriers, while the Turkish Dutch (ethnic minorities) highlighted language barriers and cultural differences as factors discouraging them from making full use of primary care in comparison to their indigenous Dutch counterparts.

There is evidence that patients in extremely poor health tend to report extreme scores either positively or negatively with regards to patient satisfaction [7]. Naturally individuals vulnerable to poor health are in a state of anxiety and stress which would affect their outlook on many aspects of life such as their consultations with their GPs.

A study of individual, practice and regional differences in patients with multimorbidity (multiple long-term illnesses) of needs not being met in the context of GP consultations and support from local services, based on the 2018 English General Practice Patient Survey (GPPS) concluded that levels of unmet needs were high particularly among specific groups that included younger patients and those from ethnic minorities [15]. It is well established that ethnic minorities are more vulnerable to a variety of long—term health problems. For instance, South Asians are at a higher risk of diabetes, coronary heart disease, asthma, gastrointestinal diseases while Black ethnic groups are at a higher risk of hypertension and diabetes [1]. Therefore, it is plausible that ethnic minorities dealing with a variety of health issues may not be having their complex needs met and report this in their patient satisfaction scores.

Good access to health services is often viewed as one of the critical elements of quality and can be categorized as a dimension of care on its own [13]. Ethnic minorities are more likely to have difficulties in effectively accessing health services due to having lower socioeconomic status. For instance, some may not be able to afford personal computers, or smart phones required to take advantage of the online booking system for GP appointments. Others may have to rely on public transport which can be an undesirable experience if their GPs are not conveniently located. If a patient has a difficult time accessing a particular health service, they may rate the whole experience negatively even if the health service provided was of good quality. However, further research is required to establish a potential link between access and overall satisfaction which remains debatable [5, 8, 13, 16–18].

A number of studies have examined ethnic differences in patient satisfaction alongside other socio-economic and demographic factors [4, 6]. The findings have confirmed the consistency of age and ethnicity as significant factors in patient satisfaction, with ethnic minorities and younger patients reporting lower satisfaction. Observed patterns were consistent with findings from analysis of the GPPS data for the years 2011–2017 which established that at least until 2017, marginalised groups (i.e. low socioeconomic status, sexual minorities, young people, and ethnic minorities) were linked to patients' low satisfaction scores [2]. A study across 31 countries in Europe further concluded that women, low-income individuals and first-generation migrants were less satisfied with their GPs [7]. These findings further raise an issue of whether a patient's satisfaction is influenced by their immigration status or their ethnicity, especially since both immigration and ethnicity account for differences in patient care experience as individuals from different countries may have different expectations of health care [8]. Less acculturated migrants may experience more barriers to health care [7]. In a study of factors accountable for ethnic differences in perceived quality of care in the Netherlands, Lamkaddem et al. [19] concluded that other than socio-demographic characteristics, cultural values had an influence on patients expectations regarding the quality of care received from

the GP. Immigrants from countries with modern and egaliteran attiudes more in line with the Netherlands were more likely to highly rate the quality of primary health care in the Netherlands.

Furthermore, an examination of barriers to presenting cancer symptoms to GPs among women in the UK identified emotional barriers as prominent among ethnic minority groups [20]. Women from minority ethnic groups were more likely to pray or prefer their traditional forms of medicine. If an individual believes that prayer and traditional medicine are better remedies for cancer symptoms in comparison to the expertise of GPs, then it is likely that they may not highly rate their experience of GP consultations since they may not fully appreciate medical science. A study in South Africa by Munyewende and Nunu [21] offers a different perspective whereby the black Africans were the ethnic majority instead of being the ethnic minority. The overall higher satisfaction (i.e. 90%) compared to the average patient satisfaction of black Africans in Europe supports the narrative that ethnicity in itself is not a strong determinant of patient satisfaction.

Munyewende and Nunu [21] further concluded that factors associated with patient satisfaction were service-related factors including time spent waiting for consultation, nurses listening skills, being given adequate information, being treated politely and having their privacy respected. It would be worth determining the importance of such factors among ethnic minorities in England or wider UK and Europe. In a study that deployed multi-level regression models on the national Quality of Life (QOF) dataset for England 2011/12, combined with the GPPS 2012 data with the aim of determining if training GP staff led to better performance on patient satisfaction scores, Ashworth et al. [22] concluded that such training did not lead to better performance under 'patient centred care' attributes such as 'Doctors listening skills' and 'Doctor's care/concern'.

From the foregoing review of factors associated with low patient satisfaction among ethnic minorities, it may be postulated that factors that contribute to certain groups reporting low patient satisfaction does not depend on ethnicity alone but on the interaction between ethnicity and other socio-demographic factors [6]. Factors identified as important include social deprivation, communication barriers, levels of acculturation, health status, health service access, GPs patient care skills, immigration status, and attitudes of immigrants towards UK health care. These factors may independently influence patient satisfaction or interact with other factors. Additionally, each ethnic group has its own distinctive history, cultural values, beliefs and unique experiences with their primary health care providers [23]. If we consider the uniqueness of different ethnic groups and the possible interrelationship between factors that may influence patient satisfaction, determining the ethnic differences in patient satisfaction becomes a complex task. This complexity may have been amplified in recent years as health care systems and characteristics unique to ethnic groups may have evolved due to the Covid-19 pandemic.

### 1.3 Aims and objectives

The preceding section highlights important gaps in existing knowledge of our understanding of ethnic inequalities in patient satisfaction in England, which warrant further research attention. In particular, there is need for an understanding of recent patterns in satisfaction of ethnic minority patients amidst Covid-19 in England, given potential implications of the pandemic for ethnic minorities, a group known to be disproportionately affected by the pandemic [24]. Furthermore, the contribution of various factors in the ethnicity and patient satisfaction relationship in England remain inconclusive from existing studies (e.g. immigration status, service access) or have not been addressed (e.g. role of internet access and use of online services which has become particularly crucial during Covid-19). While a number of studies in England have recognized the importance of multilevel determinants of patient satisfaction and

adopted multilevel regression approaches, most attention has been on the relationship between ethnicity and patient satisfaction at individual level, focussing on demand-side factors based on patient characteristics. Whether observed patterns in the relationship would hold at aggregate General Practice (GP) surgery and Clinical Commissioning Group (CCG) level are unclear. An improved understanding of ethnic inequalities at GP and CCG levels is required to inform system level decisions at GP and CCG levels by relevant policy holders and planners within the health care ministry to effectively enact system level policies while considering patient experience. With each GP practice being part of a CCG which was considered the cornerstone of the health system in England [25], there is need for improved understanding of variations at both GP and CCG level. The current study aims to contribute to these existing gaps in knowledge.

The main objective of this paper is to gain a better understanding of factors that contribute to ethnic inequalities in overall patient satisfaction with their primary health care in England. The specific objectives are to (i) examine recent ethnic inequalities in patient satisfaction with primary health care in England at GP and CCG levels; and (iii) establish factors that contribute to ethnic differences in patient satisfaction, with particular focus on service-related factors.

## 2: Materials and methods

### 2.1 The Data and variables

This paper is based on secondary analysis of the General Practitioner Patient Survey (GPPS) datasets. The GPPS is nationally-representative repeated cross-sectional survey in England. It is one of the largest surveys of millions of registered NHS patients invited to disclose their experience in relation to the primary care they receive [26]. The patients must have been registered with a General Practice (GP) for at least 6 months. It is an independent survey executed by IPOS on behalf of NHS England and is conducted annually. The large sample size of the GPPS and national representativeness makes it ideal for this study, providing results that can be generalized to the population of England.

The GPPS dataset contains individual-level data as well as data aggregated at GP surgery level whereby each case represents a GP surgery. This study is based on GP-level data. The GPPS datasets are available to the interested public/researchers and free to download under 'UK open data' (https://gp-patient.co.uk/surveysandreports). We analysed datasets for three recent surveys undertaken in 2019, 2020 and 2021 (see Table 1 below).

Key study variables included variables on ethnicity, patient satisfaction and a list of variables associated with ethnicity and/or patient satisfaction that could explain/modify any observed relationship between ethnicity and patient satisfaction. The list of GPPS variables extracted for analysis are included in S1 File. From the list of variables in S1 File, some of the variables were excluded from the final model following further data processing.

### 2.2 Methods of data analysis

Descriptive analysis of ethnic inequalities in patient satisfaction at both GP and CCG level across recent years, was followed with multilevel modelling to establish service-related factors contributing to the observed ethnic inequalities in patient satisfaction.

**Table 1. General information on the 2019,2020 and 2021 GPPS datasets for England used in the study.**

| Dataset | Fieldwork dates | No of GP surgeries (cases) participated |
|---|---|---|
| GPPS 2019 | January—March 2019 | 6999 |
| GPPS 2020 | January—March 2020 | 6821 |
| GPPS 2021 | January—March 2021 | 6658 |

**2.2.1 Descriptive analysis**. Descriptive analysis was undertaken to examine ethnic differences in patient satisfaction across recent three years (2019, 2020 and 2021). This involved first running univariate analysis on each of the key study variables for ethnicity and patient satisfaction to understand their distributions before carrying out appropriate bivariate analysis to understand the associations. Since all variables at aggregate GP level were interval-level (continuous variables) based on percentages or proportions, univariate analysis was based on descriptive statistics and distributions (histogram-normality), the latter being particularly important for the outcome variable on patient satisfaction to enable assessment of appropriateness of distribution for regression analysis.

Univariate analysis was followed with bivariate analysis to understand recent patterns in association between patient satisfaction and ethnicity. This involved use of correlation analysis and scatter plots to understand the association between the proportion of patients in GP-practice or within CCG who reported overall good (very good or fairly good) experience and proportions of patients of different ethnic groups. To enable a clearer visualization of the association between ethnicity and patient satisfaction, key variables for patient satisfaction and ethnicity were classified into categorical variables (based on quartiles), to enable clearer graphical presentation (stacked bar charts) and cross-tabulation (with Chi-Square test) used to further assess the relationship.

**2.2.2 Multilevel regression analysis.** A two-level regression analysis, with CCG as level-2 and GP as Level-1, was used to examine the relationship between patient satisfaction and ethnicity, taking into account other factors that may have an influence on the relationship. A multilevel linear regression model was considered appropriate to control for potential correlation of GPs within each CCG, since ignoring such correlation would violate underlying regression analysis assumptions [27]. Furthermore, we were interested in establishing the extent to which patient satisfaction with their GP service was attributable to CCG level factors, especially since CCG level policies are likely to influence GP practices which may have implications for patient satisfaction of ethnic minorities within specific CCGs. The outcome or dependent variable was interval-level, measured as the proportion of patients in a GP surgery who rated their satisfaction as very/fairly good, while the proportion of ethnic minorities was the key independent variable. All the other variables were set as covariates and introduced in the model in successive stages to determine the extent to which they explained the net effect of ethnicity on overall patient satisfaction. The two-level linear regression model used was of the form:

$$Y_{ij} = B_0 + B_1 X_{1ij} + \ldots + B_k X_{kij} + u_j, + e_{ij}$$

where $Y_{ij}$ is patient satisfaction (proportion with good overall satisfaction) for GP $i$ in CCG $j$; $B_0$ is the regression constant/intercept (average patient satisfaction when all independent variables are zero); $X_{1ij}$- $X_{kij}$ are ethnicity and other independent variables considered in the model which may be defined at GP of CCG level; and $B_1$-$B_k$ are the associated parameter estimates. The quantities $u_j$ and $e_{ij}$ are the residuals at CCG and GP levels, and assumed to have normal distributions with mean of zero and variances $\sigma^2_u$ and $\sigma^2_e$. respectively [27].

A useful measure for examining the extent to which GP patient satisfaction varies across CCGs, intra-unit correlation, $\rho_u$, is derived as the ratio of the variance at CCG level to the total variance, defined as:

$$\rho_u = \sigma^2_u / (\sigma^2_u + \sigma^2_{e).}$$

For the multilevel analysis, careful preliminary analysis was undertaken to determine the key variables that would be used in the final model. A study by Hudson and Smith [17] assessed the influence of different measurements of service quality on patient experience in

primary healthcare and noted that to improve patient experience, focus should be placed of responsiveness of practise and interactions with the physician. Considering this, from the range of variables included in the GPPS, those listed in S1 File were considered to be key variables relevant for determining patient experience that need to be considered in the multilevel analysis. Part of the preliminary analysis for the regression analysis involved determining which explanatory/independent variables could be simultaneously including in the model, since including highly correlated variables could lead to multicollinearity [28] and was therefore avoided. Correlation analysis between all variables considered as explanatory variables was undertaken to determine their correlations and variables with particularly high correlation (r>0.8) removed accordingly. Further preliminary analysis was undertaken to inform effective grouping of specific variables included in the model. A breakdown of the ethnic groups in the GPPS data obtained during descriptive analysis on the variables for the different ethnicity groups showed that the percent of individual ethnic minorities (non-white) were too small for them to be considered on their own for the regression analysis. Therefore, all the ethnic minorities were combined into one group for the multilevel analysis. Similarly, all White ethnic groups (including those from EU and other countries) were grouped together due to small numbers. SPSS (Statistical Package for Social Sciences) analytical tool was used to perform descriptive analysis and graphical presentation. Multilevel analysis was undertaken using MLwiN [29].

**2.2.3 Ethical considerations.** We wish to declare that research reported in this article meets the standard ethical requirements. The study is based on secondary analysis of the GP Patient Survey which has been designed to give patients the opportunity to give feedback about their experiences of their GP practice. It is carried out by Ipsos MORI, on behalf of NHS England. Ipsos MORI is a registered and independent survey organization that strictly adheres to the Market Research Society's ethical code of conduct (http://www.gp-patient.co.uk/faq). Furthermore, the dataset analysed in this paper was already aggregated at GP level, so it is impossible to link data to a particular individual. Ethics approval was not required for this study since we used anonymised data aggregated at the GP level and routinely available in the public domain.

# 3: Findings

This section presents findings of analysis undertaken to address the specific research objective to: (i) examine recent ethnic inequalities in patient satisfaction with primary health care in England at GP and CCG levels;; and (iii) establish service-related factors that contribute to ethnic differences in patient satisfaction. The first objective is addressed through descriptive univariate and bivariate analyses, while the second objective is addressed through multilevel modelling.

## 3.1 Descriptive analysis of ethnic inequalities in patient satisfaction in England

Descriptive analysis started with univariate analysis to understand the distribution of key study variables relating to ethnicity and patient satisfaction before examining the association between the two using bivariate analysis. The proportion of respondents of different ethnic groups across GP surgeries in England presented in Table 2 shows that patients of white ethnicity make up the vast majority of respondents of about 82% as may be expected. The proportion of white patients across GP practices ranges from 0% to 100%. Overall proportion of ethnic minority respondents at GP practices in England ranges from 0% to 100%, with a mean of about 17–18%. Asians make up the largest proportion of ethnic minority respondents (10–

**Table 2. Proportion of respondents at GP practice of different ethnicity in England (2019, 2020 and 2021).**

| Year | Ethnicity | N (GPs) | Minimum | Maximum | Mean | Std. Deviation |
|------|-----------|---------|---------|---------|------|----------------|
| 2019 | White | 6304 | .00 | 1.00 | .8222 | .22261 |
|      | Mixed | 6528 | .00 | .27 | .0159 | .02208 |
|      | Indian | 6760 | .00 | .93 | .0370 | .08578 |
|      | Pakistani | 6846 | .00 | .86 | .0283 | .08785 |
|      | Bangladeshi | 6880 | .00 | .66 | .0106 | .04196 |
|      | Other Asian | 6622 | .00 | .81 | .0271 | .04376 |
|      | All Asian | 6357 | .00 | .96 | .1015 | .16600 |
|      | Black | 6655 | .00 | .60 | .0394 | .07092 |
|      | Arab and other | 6718 | .00 | .68 | .0251 | .04164 |
|      | All minority | 5666 | .00 | 1.00 | .1698 | .22118 |
|      | Valid N (GPs) | 5235 | | | | |
| 2020 | White | 6249 | .00 | 1.00 | .8179 | .22619 |
|      | Mixed | 6377 | .00 | .19 | .0168 | .02179 |
|      | Indian | 6595 | .00 | .94 | .0376 | .08604 |
|      | Pakistani | 6667 | .00 | .89 | .0289 | .08934 |
|      | Bangladeshi | 6692 | .00 | .78 | .0105 | .04095 |
|      | Other Asian | 6450 | .00 | .75 | .0279 | .04394 |
|      | All Asian | 6228 | .00 | .97 | .1022 | .16437 |
|      | Black | 6475 | .00 | .62 | .0386 | .06791 |
|      | Arab and other | 6521 | .00 | .73 | .0258 | .04286 |
|      | All minority | 5566 | .00 | 1.00 | .1716 | .22157 |
|      | Valid N (GPs) | 5215 | | | | |
| 2021 | White | 5790 | .01 | 1.00 | .8056 | .22923 |
|      | Mixed | 5923 | .00 | .15 | .0172 | .02069 |
|      | Indian | 6337 | .00 | .91 | .0414 | .09013 |
|      | Pakistani | 6465 | .00 | .81 | .0287 | .08505 |
|      | Bangladeshi | 6510 | .00 | .75 | .0115 | .04327 |
|      | Other Asian | 6073 | .00 | .77 | .0300 | .04596 |
|      | All Asian | 5671 | .00 | .94 | .1103 | .17152 |
|      | Black | 6072 | .00 | .58 | .0402 | .06803 |
|      | Arab and other | 6220 | .00 | .63 | .0262 | .04082 |
|      | All minority | 4531 | .00 | 1.00 | .1780 | .22697 |
|      | Valid N (GPs) | 4046 | | | | |

11% of all respondents), followed by Black ethnicity (about 4%) and Mixed (1.6–1.7%). These ethnic distribution patterns are more or less consistent with estimates for the UK overall population, although overall UK estimates suggested a slightly higher proportion of Whites (86%) and slightly lower proportions of Asians (7.2%) and Blacks (3.2%) in 2016 [30].

Analysis of the distribution of the main outcome variable on patient satisfaction (percent reporting very good or fairly good overall experience) shows that overall satisfaction has remained consistently high during the last three years. Although the satisfaction levels seem fairly consistent across the three years, with 82–83% of the respondents reporting overall good experience, the non-overlap in 95% confidence intervals suggests significant differences [28], with satisfaction level in 2020 (82.5%, CI: 82.2–82.7%) being significantly lower than either 2019 (83.4%, CI: 83.1–83.6%) or 2021 (83.5%, CI: 83.2–83.7%). However, there is no evidence of declining or increasing trends in patient satisfaction during this period since there is no significant difference between 2019 and 2021.

The results of bivariate analysis based on correlation, undertaken to understand the relationship between ethnicity and patient satisfaction across years is presented in S2 File. The correlations between ethnicity and patient satisfaction are fairly consistent across the three years, with a significant positive correlation between white ethnicity and patient satisfaction, and significant negative correlation between patient satisfaction and all ethnic minority groups. Although the correlations are not particularly strong (r<|0.4|), they are all highly significant (p<0.001) across all the three years. Among all the ethnic minority groups, Asians have the strongest negative correlation, and among the Asians, the Pakistani have the strongest negative correlation, suggesting that GP surgeries with the highest proportion of Asian/Pakistani respondents were the least likely to report good overall experience. The correlation of ethnicity and patient satisfaction at CCG level based on the 2021 dataset (not shown) gives an even stronger relationship, suggesting that CCGs with higher percentages of ethnic minority patients have significantly lower patient satisfaction (r = -0.49, p<0.001).

Overall patient satisfaction by ethnicity is more clearly presented in the stacked bar charts in Fig 1 (based on 2021 data for illustration), obtained after classifying both ethnic minority and patient satisfaction proportions by quartiles.

The relationship between overall patient satisfaction scores and the combined ethnic minority groups depicts a clear negative relationship between minority ethnic group and patient satisfaction (Chi-Square p-value<0.001). As the proportion of ethnic minorities (All non-white groups) within a surgery increases, the patient satisfaction scores within that surgery decreases. For instance, the surgeries in lowest quartile of ethnic minorities had about 43% of respondents reporting the top most 25% satisfaction. On the contrary, less than 10% of surgeries in the highest quartile of ethnic minorities reported the top most 25% satisfaction. Similar and even stronger patterns were observed at CCG level based on the 2021 dataset which showed that: the CCGs in lowest quartile of ethnic minorities had about 46% of

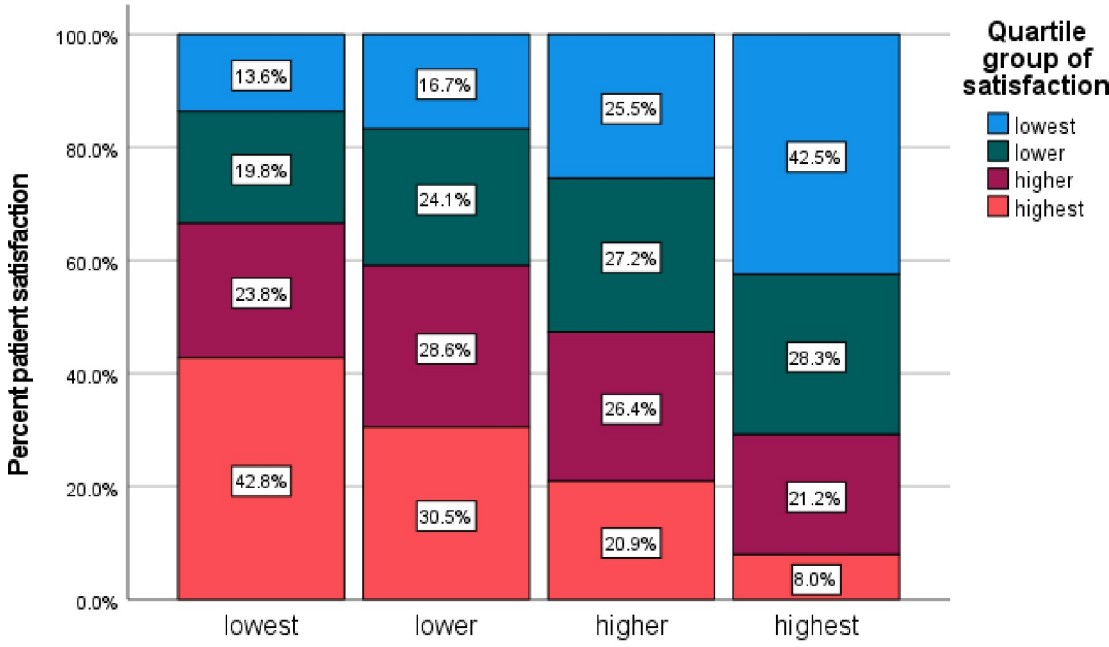

**Fig 1. Distribution of patient satisfaction by ethnicity (England, 2021).**

respondents reporting the top most 25% satisfaction. On the contrary, less than 5% of CCGs in the highest quartile of ethnic minorities reported the top most 25% satisfaction.

## 3.2 Multilevel analysis of factors that contribute to ethnic differences in patient satisfaction

Multilevel linear regression was used to enable a deeper understanding of the relationship between patient satisfaction and ethnicity. This was preceded with preliminary analysis to guide the model building/selection process and interpretation of regression analysis results. Based on descriptive analysis in the preceding section, it was necessary to use the broader classification of ethnicity of White/Minority. Since there was no evidence of significant trends in patient satisfaction by ethnicity, the multilevel analysis was based on the most recent data at the time of analysis (i.e 2021). Some of the basic underlying assumptions of linear regression analysis, including normality and linearity [28], require a good understanding of the distribution of the dependent variable as well as relationship with explanatory variables to be included in the model.

Linear regression diagnostics (not shown) suggested that none of the basic underlying assumptions (normality, homogeneity of variance and linearity) were seriously violated. Although there was evidence of mild violation of the normality assumption, results are expected to be fairly robust with mild-moderate violation of normality assumption where sample sizes are large [28], as in this case.

Further preliminary analysis involved use of scatter plots to better understand the nature of the relationship between patient satisfaction and ethnicity, at both GP and CCG levels for 2021 (Figs 2 and 3). Besides confirming the negative association between patient satisfaction with minority ethnicity, Figs 2 and 3 provide useful information on the nature of the distribution in the relationship, with most observations clustered towards the end of lowest minority

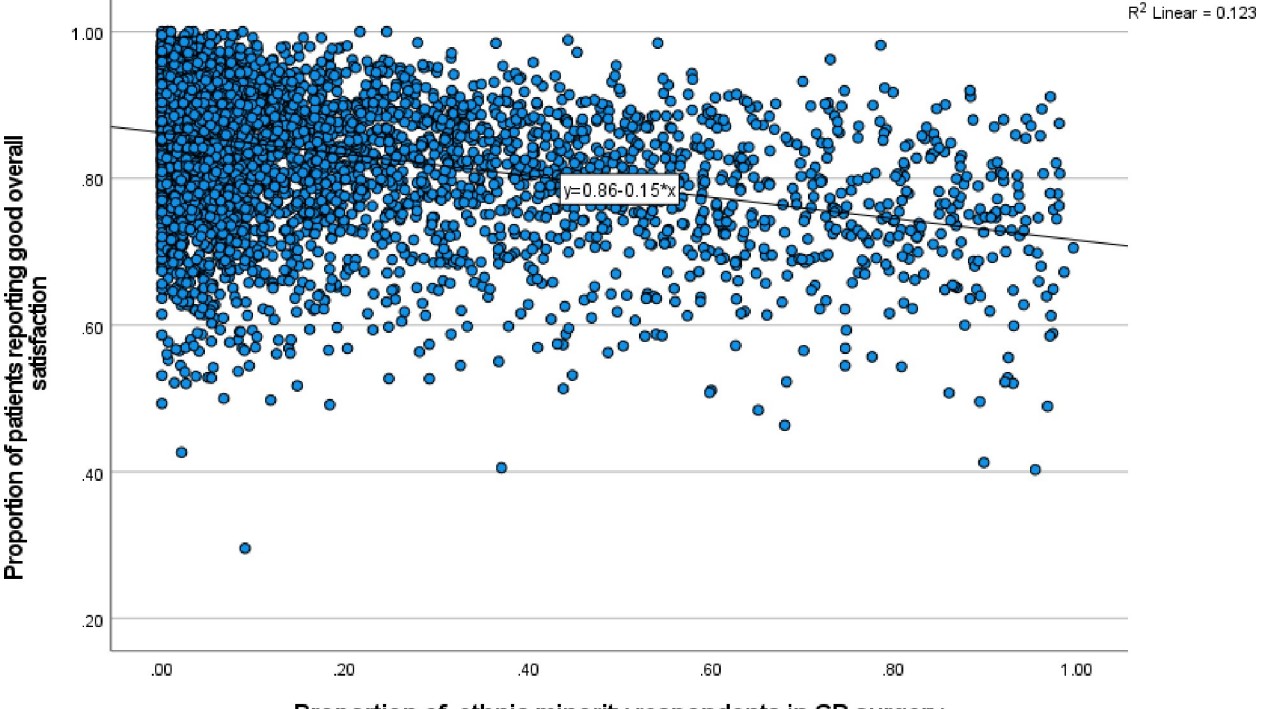

**Fig 2. Scatter plot of overall satisfaction vs minority ethnic group at GP level.**

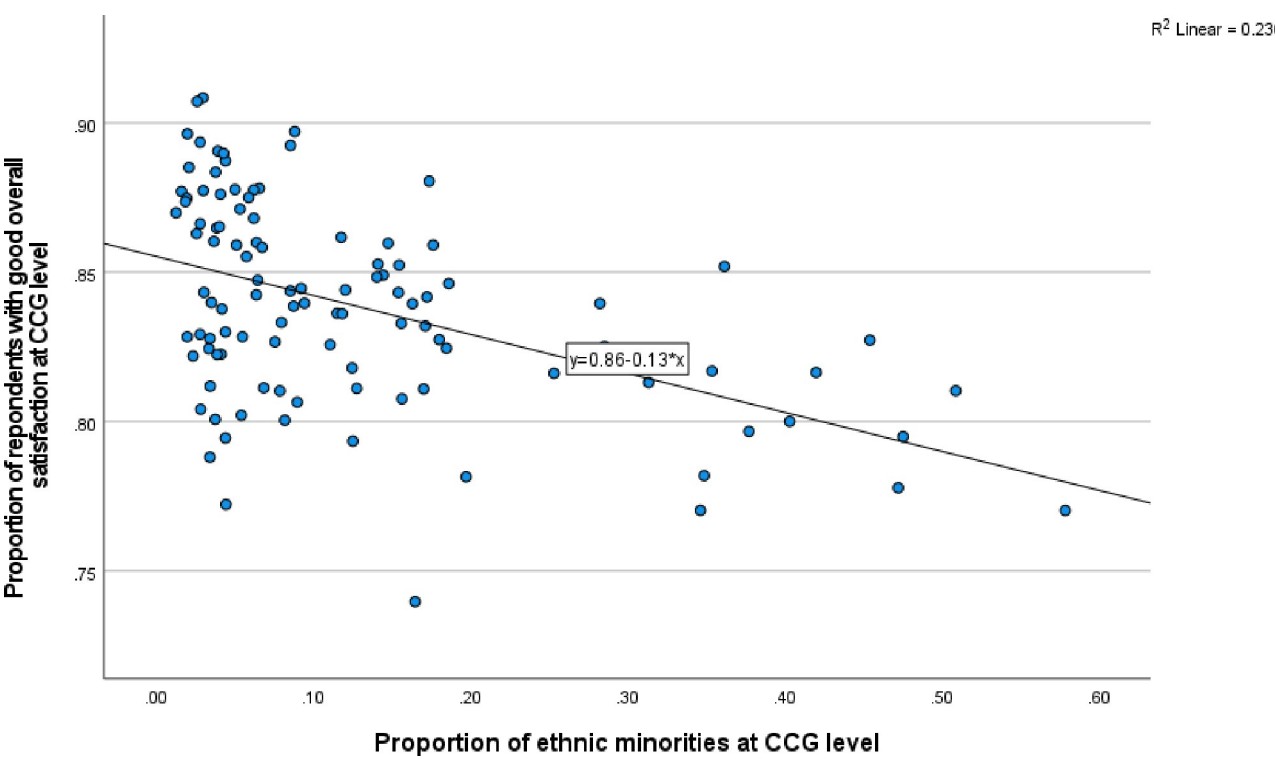

**Fig 3. Scatter plot of overall satisfaction vs minority ethnic group at CCG level.**

ethnicity. Despite the uneven distribution, there is no evidence of the relationship being non-linear. From the equations of the two simple liner regression lines, it is clear that there is a significant linear relationship at both GP and CCG levels. Indeed, the relationship is stronger at CCG level, further supporting the application of multilevel analysis to control for potential correlations at CCG level. While the percent of variation in patient satisfaction attributable to minority ethnicity is about 12% at GP level, this is about double (24%) at CCG level (based on R-square).

The multilevel model building process started with a variance components model with no explanatory variables to examine variations in ethnic satisfaction across CCGs (Model 0). Results showed a highly significant variation in patient satisfaction across CCGs, with intra-class correlation coefficient (ICC) of 11%. This suggests that 11% of the total variation in GP patient satisfaction (i.e. proportion of patients in GP unit rating their overall satisfaction as fairly/very good) is attributable to CCG level factors while the remaining 89% is attributable to GP level factors.

Multilevel linear regression started with ethnic minority as the key independent variable and patient satisfaction as the outcome (Model 1) before introducing each of the factors that are expected to explain the effect of ethnicity on satisfaction to the model, one at a time, and noting changes in the effect of ethnicity. The modelling considered both demand side factors relating to patient characteristics that have been identified in the literature to mediate or moderate the relationship (e.g long-term health condition, age, and working status—considered as a proxy for socio-economic status), as well as supply side factors relating to quality of care received. Results of significant factors are presented in Table 3.

Results for Model 1 with only ethnicity included in the model confirm that minority ethnicity is associated with significantly lower patient satisfaction. Across all CCGs, the predicted

Table 3. Multilevel Linear regression results of predictors of patient satisfaction (England 2021).

| Factor | Model 0 | Model 1 | Model 2 | Model 3 |
|---|---|---|---|---|
| Constant | 0.84 (0.0033) | 0.86 (0.0032) | 0.95 (0.0111) | -0.36 (0.0236) |
| Minority ethnic (%) | | -0.15 (0.0075)** | -0.17 (0.0082)** | 0.00 (0.0049) |
| Long-term Health condition | | | -0.15 (0.0188)** | 0.00 (0.0106) |
| Ease of using website | | | | 0.22 (0.0081)** |
| Always seen preferred GP | | | | 0.12 (0.0046)** |
| Care and concern | | | | 0.61 (0.0190)** |
| Confidence of managing condition | | | | 0.05 (0.0109)** |
| Trust in healthcare professional | | | | 0.23 (0.0324)** |
| Involved in decisions about treatment | | | | 0.18 (0.0428)** |
| *Random effects* | | | | |
| CCG variance (S.E) (10 EXP-4) | 0.98 (0.1593) ** | 0.71 (0.1305)** | 0.67 (0.1240) ** | 0.04 (0.1301)** |
| ICC | 0.111 | 0.089 | 0.085 | 0.019 |

*significant at 5% level (p<0.05)

**significant at 1% level (p<0.01)

average satisfaction for a GP surgery with no ethnic minority patient (0% ethnic minority) is 86%. The regression coefficient of -0.15 implies that a GP surgery with 100% ethnic minorities would have a predicted average satisfaction of 71% (i.e. 0.86–0.15), before controlling for any other predictors.

Of all the demand side factors relating to patient characteristics considered in the model, only long-term health condition was observed to be significant. Although bivariate correlations (not shown) between long standing health condition and patient satisfaction suggested a positive relationship, the results of Model 2 (after adding long-term health to the model) suggest that once ethnicity is accounted for, long-standing health condition is associated with lower patient satisfaction. The results for Model 2 further suggest that introducing long-standing heath condition to the model enhances (rather than diminishes) the negative association between minority ethnicity and patient satisfaction.

A number of supply side factors relating to service provided at GP surgery were significant predictors of patient satisfaction and largely explained the observed lower satisfaction among ethnic minority respondents (Model 3). In particular, ease of using GP practice's website to look for information or access services was associated with a significant increase in patient satisfaction and a notable drop in the negative relationship between ethnic minority and patient satisfaction (See S2 File for stepwise results). This suggests that the observed lower satisfaction among minority ethnic groups was partly due to difficulties in using their GP practice's website to look for information or access services. Although always seeing preferred GP was a significant predictor of patient satisfaction, introducing this variable in the model resulted in minimal reduction in the effect of ethnicity, suggesting that it was not a particularly important factor in explaining the lower satisfaction among ethnic minority patients. Being treated with care and concern was a strong predictor of patient satisfaction, and explained a substantial proportion of ethnicity effect (See S2 File). As expected, being treated with care and concern was associated with significantly higher satisfaction, and was an important factor in explaining the lower satisfaction among ethnic minority groups. Confidence of managing condition was a significant factor in patient satisfaction and explained most of the effect of long-term health condition, and some ethnicity effect. Finally, both trust in healthcare professional and involvement in decisions about treatment were associated with higher patient satisfaction, and together with the factors outlined above, fully accounted for the ethnicity effect in patient

satisfaction. These GP service factors also largely explained the observed CCG level variations, as shown by the reduction in ICC in Model 3. Once these were controlled for, only about 2% of the total unexplained variation in patient satisfaction was attributable to unobserved CCG level factors.

In addition to the factors presented in Table 3 above, the multilevel regression analysis considered a number of factors relating to patient characteristics including age and working status, identified in the literature to moderate/mediate the relationship between ethnicity and patient satisfaction. There was no evidence that the main effect of these factors (e.g percent aged 65+, percent in paid FT/PT work, etc), nor their interactions with ethnicity were significant. Therefore, these factors were not included in the models presented.

## 4: Discussion

The findings presented in this paper on ethnic disparities in patient satisfaction are generally consistent with results from previous studies, while findings on factors contributing to the observed ethnic difference provide important new insights. The finding that GP surgeries with higher percentages of ethnic minorities have a lower overall satisfaction score indicates that ethnic minorities in general report lower patient satisfaction. This is widely supported by most if not all of previous research with similar objectives [2, 4, 6, 13], suggesting that aggregate (GP-level) associations (in this study) are consistent with patterns observed at individual patient level. However, the reasons for the lower patient satisfaction score among ethnic minorities is yet to be fully determined and is where the main contribution of this study lies.

In our predictive analysis based on multilevel linear regression modelling, we considered a range of factors, including patient characteristics and service-related factors that were presumed to be potentially important based on previous research/literature. We identified one demand/patient related factor (long term health condition) and a number of supply/service related factors (ease of using website; always being able to see preferred GP; being treated with care and concern; confidence of managing health condition; trust in Health care professional; and being involved in decisions about treatment) as significant factors that account for observed ethnic differences in patient satisfaction at GP level.

### 4.1 Demand/Patient related factors

Long term health status was the only significant patient-related factor in our analysis. Previous studies suggest that patients' long term health status is likely to affect their satisfaction with care. For instance, Detollenaere et al. [7] noted that patients with long term health issues are likely to score either extremely positively or negatively in patience satisfaction as they are likely to be either appreciative or frustrated with their constant interactions with the primary care providers. Since ethnic minorities are more vulnerable to long- term health issues [31], it was important to establish the extent to which long-term health issues may have been a factor in ethnic disparities in patient satisfaction. When we introduced the variable for long term health condition into the model, it had an effect on the impact of ethnicity in the overall model. Although we had expected the impact of ethnicity to reduce with the introduction of long-term health status, it increased instead. Investigating further, we ran a correlation analysis of long-term health status and ethnic minority variables. The result was a significantly high negative correlation, suggesting that the ethnic minorities who participated in the study were less likely to report having a long-term health condition. When we run correlation analysis between the different age groups and long-term health status, the older age groups had a significant positive correlation with long term health status. This was expected as older patients are most likely to report long term health conditions. A correlation analysis between Ethnic

minority and the different age groups indicated that ethnic minorities who participated in the study were relatively younger than their White counterparts. Therefore, it was apparent that older patients were more likely to have a long health term condition, while ethnic minorities were predominantly made up of younger age groups. Long term health status did not account for ethnic differences in overall satisfaction but when factored in it did affect the relationship between ethnicity and overall satisfaction. This did not support our hypothesis that Ethnic minorities would be susceptible to long term health issues, and more likely rate their overall satisfaction more extremely, either positively or negatively, depending on whether they are appreciative or frustrated with their health care.

We considered introducing several other variables in the model, but they were not significant and therefore excluded. Among these variables were variable pertaining to age and working status. In previous studies age was determined to be a factor in ethnic differences of patient satisfaction but there was no evidence to support this in our analysis. As noted in the literature review, the relationship between ethnicity and patient satisfaction can be moderated or mediated by a range of demographic factors, leading to rather complex patterns. Interactions between various socio-economic (e.g percent in paid work) and demographic factors (e.g percent aged 65+) with ethnicity were considered in the regression model but there was no evidence that any of these were significant.

## 4.2 Supply/Service related factors

The (Ease of using Website) variable was included as a covariate in our multilevel regression analysis as it was one of the few viable variables that could be used as a proxy for access to health services. As noted by Kontopantelis et al. [13], good access is critical to good health care. Our hypothesis was that ethnic minorities would lack good access to health care, presumably due to lack of financial resources [32] and issues of access and language barriers which may affect their overall patient satisfaction. Individuals with access to devices such as laptops, high spec smart phones, tablets etc will be able to receive the maximum benefits of the online services available which is likely to result in a better overall patient experience and satisfaction. A larger percentage of ethnic minorities fall within the lower socio-economic group and may not be able to afford such devices. As expected, when we introduced the (Ease of using Website) variable into our multilevel regression model, the effect of ethnicity on overall satisfaction decreased substantially, suggesting that service accessibility was an important factor in partly explaining the lower satisfaction among ethnic minority patients.

We had stipulated that ethnic minorities, especially first-generation immigrants, would feel more comfortable seeing a GP who can speak the same language and share the same cultural norm, especially since existing studies such as Burt et al. [6] and Schinkel et al. [14] highlight communication and acculturation as critical factors in ethnic differences in patient satisfaction. When the (Always seen preferred GP) variable was added to the multilevel regression model, the effect of ethnicity on overall satisfaction further decreased as expected, although the decrease was marginal. We had further hypothesized that ethnic minorities would be more likely to negatively rate the 'Care and concern' they received partly due to cultural differences and communication barriers that would lead to misunderstandings and frustrations between patients and the GP staff. When the (Care and concern) variable was added to the model, the effect of ethnicity on overall satisfaction further decreased as expected. The decrease was substantial, suggesting that this was an important factor in explaining the overall low satisfaction among ethnic minorities.

An earlier study conducted by Williams et al. [20] to determine ethnic differences in barriers to presentation of cancer symptoms in primary care among women in England concluded

that ethnic minority groups were more likely to pray about a symptom or rely on traditional remedies. Some ethnic minorities, especially first-generation immigrants, are more likely to rely on traditional forms of medicine or prayer instead of the primary care offered as a solution to health issues. They would be more confident in self managing their long-term conditions and may not appreciate the primary care offered nor have trust in health care professionals. Therefore, we had hypothesized that ethnic minorities would be more confident in self managing their condition (through prayer and traditional medicine) and have little trust in health care professionals that would result in lower overall satisfaction. When the variables (Confidence in managing condition) and (Trust In health care Professional) were introduced in the model, there was a decrease in the effect of ethnicity in overall satisfaction as expected, but the decrease was minimal.

Schinkel et al. [14] conducted a study in the Netherlands to determine the perception among Turkish—Dutch migrants in relation to barriers in their involvement in primary health care consultations. The study noted that the Turkish—Dutch expected a larger power distance in medical consultations, they expected the doctor to take full control while they remained passive as they were acclimated to in Turkey. Some ethnic minorities especially immigrants from non–western cultures, may have a preference of a certain power dynamic in the doctor-patient relationship. They may prefer the doctor to assume a dominant role and take full control of the consultation, and do not expect nor desire to be involved in decisions about the treatment as they expected the doctor to know everything and dictate the course of action. This expected doctor—patient relationship is not in line with practices in developed western countries where doctors are encouraged to involve the patient as much as possible while determining the course of treatment. We had hypothesized that some ethnic minorities may rate the overall satisfaction more negatively if they felt pressured to get more involved in the consultation process as they may not be accustomed to it. When the (Involved in decisions about treatment) variable was added to the multiple regression analysis, the change on effect of ethnicity on overall satisfaction was minimal.

### 4.3 Strengths and limitations

It is important to recognize some data limitations that may have potential implications on interpretation of the findings presented in this paper. Although use of secondary data analysis has many advantages (including availability of large-scale national datasets that would yield nationally representative results; data already anonymised to minimize ethical concerns, etc), there are clear limitations as well [33]. First and foremost, some of the variables of interest pertaining to racism or variables pertaining to immigration status are not available in the dataset. Understandably racism whilst receiving primary care is extremely difficult to reliably capture and measure using a questionnaire.

Also important are possible reporting biases. In a comprehensive study by Burt et al. [5] to determine how different patients use services to record their primary care experience, one of the conclusions was that patients would readily criticize a consultation they viewed on video but would be hesitant to criticize a consultation on a questionnaire. As our analysis is based on data gathered from questionnaires, it is important to consider whether analysing data gathered from other media would have provided more reliable results, especially for ethnic minorities. A study by Burt et al. [6] investigating differences in GP–Patient communication by ethnicity, age, gender, further noted that if survey's respondents are typically proficient in English then difficulties in understanding the questionnaire that are likely to be experienced among some of the ethnic minorities may be overlooked. Since this paper focuses on patient satisfaction of ethnic minorities, it is possible that if respondents among these groups did not fully

understand the questionnaire, they may provide invalid responses yielding invalid data for the analysis.

Furthermore, our analysis is based on data limited to England, so it is likely that our findings may only be applicable to the health care system in England. Munyewende and Nunu [21] while conducting a study on patient satisfaction in nursing delivery in South Africa, highlighted that different countries have different contextual settings and there is a lack of standardized methodologies in measuring patient satisfaction across countries. Measuring patient satisfaction across different countries/continents is a challenge and therefore efforts to standardize the methods and approaches would improve generalizability of findings to other regions.

Finally, it is important to note that the analyses presented in this paper are based on proportions of respondents from ethnic groups in GP surgeries rather than individual level ethnicity which has its merits and limitations. Although use of aggregate-level group proportions or means in regression analysis has the advantage of avoiding extreme bias in parameter estimates, limitations include loss of information at individual-level and reduction of variability in the data due to aggregation [34]. Nevertheless, use of aggregate-level proportions is considered appropriate in this case since the main focus of this paper is to advance understanding of ethnic inequalities in patient satisfaction at GP and CCG levels, an area that has received limited attention in previous research.

## 5: Conclusions and recommendations

### 5.1 Summary and conclusions

This paper set out to improve understanding of factors that contribute to ethnic differences in patient satisfaction in England. The specific objectives were to: examine ethnic differences in patient satisfaction with primary care in England at GP and CCG level; and establish factors that contribute to ethnic differences in patient satisfaction in recent years. The findings, based on bivariate analysis of recent General Practitioner Patient Survey (GPPS) confirm a significant negative association between patient satisfaction and each of the main ethnic minority groups in England. Further examination of the distribution of patient satisfaction by ethnicity, based on combined ethnic minority groups, depicted a clear negative association between ethnic minority group and patient satisfaction. Observed patterns suggest that the findings presented here based on analysis of aggregate (GP surgery) level data are consistent with patterns observed from previous studies based on individual-level data.

An examination of recent patterns based on 2019, 2020 and 2021 datasets suggest that the ethnic differences in patient satisfaction have persisted in recent years. Observed patterns provide no evidence that ethnic differences in patient satisfaction have changed in recent years due to Covid-19, despite the disproportionate impact of the pandemic on ethnic minorities in England.

Multilevel regression analysis identified a number of service-related factors at GP surgery that explain the observed ethnic disparities. Significant factors included: ease of using GP website; frequency of seeing preferred GP; being treated with care and concern; confidence of managing condition; trust in Health care professional; and being involved in decisions about treatment. Two factors in particular (ease of using GP website and being treated with care and concern) largely explained the observed ethnic differences in patient satisfaction. Of all factors relating to patient characteristics considered in the analysis, only long-term health condition was significant, and it enhanced rather than reduced the negative experience of ethnic minorities. The conclusion from these findings is that existing ethnic differences in patient satisfaction at GP level is largely attributable to supply (service related) rather than demand (patient

characteristics) factors. These findings have important implications for health care system policy and practice in England, at both GP and CCG levels.

Overall, the findings confirm most of the key results from existing literature on ethnic disparities, strengthening the theoretical base of the evidence on ethnic differences in patient satisfaction. In addition, the study has provided important new insights, making an important contribution to improved understanding of service-related factors that explain existing ethnic differences in patient satisfaction.

## 5.2 Recommendations for policy and practice

The findings relating to factors that contribute to ethnic differences in patient satisfaction in England have important implications for health care system policy and practice. As identified in the multilevel analysis, the factors established to have the most significant impact in observed ethnic disparities include: ease of using GP website and being treated with care and concern. Therefore, it would be important to consider policies in relation to these factors.

Ease of using GP website relates to issues of access and language barriers. To access the GP website, patients would need access to relevant devices such as laptops, smart phones and tablets. A disproportionately large proportion of patients belonging to minority ethnic groups lack financial resources to enable them afford such devices. The GP surgeries (supported by relevant CCGs) could consider installing hubs with computer screens at surgeries that would be available to patients who needs to access the GP website. To address the issue of language barriers, there is need to work towards ensuring key information on the GP website is translated into several languages, especially ones that would cater for the most affected ethnic groups. As noted in this analysis, the Pakistani ethnic group is the most negatively affected with regards to overall satisfaction.

Communication and acculturation factors have been the source of many issues faced by ethnic minority groups whilst receiving primary care. Perceiving to not being treated with care and concern is one of the key issues, and likely to be a result of misunderstandings or miscommunication between minority ethnic patients and their GPs. The NHS could work towards implementing or reviewing the cultural competency component in the training of health care professionals. This could help mitigate the differences of cultural norms in the context of medical consultations leading to better communication and understanding between ethnic minority patients and their physicians. Furthermore, allowing more time for GP consultation may also help address this, enabling GPs to spend more time with their patients to better understand their concerns.

## 5.3 Recommendations for further research

One of the main objectives of this paper was to examine recent patterns in ethnic disparities in patient satisfaction to understand the potential impact of the Covid-19 pandemic. Future studies should build on this to better understand how ethnic minority patients may be uniquely affected, given the disproportionate impact of the pandemic on this population sub-group. The dataset used in the current study had fieldwork carried out from January to March 2019, 2020 and 2021. This timing may have failed to capture the respondents during the peak of the Covid-19 pandemic, hence it is likely that the effect of the Covid-19 pandemic [35] may not have been adequately captured and require further research attention as more recent data become available.

Finally, a better understanding of how the factors identified to contribute to negative experiences of ethnic minorities operate could be better established through qualitative research. There is need for follow-up qualitative research among ethnic minorities to have an in-depth investigation of their experiences in relation to difficulties accessing information from the

website and feelings about how they are treated by their GPs. This would enable formulation/ development of relevant policies /interventions to effectively address factors that contribute to negative experience of ethnic minority patients.

## Supporting information

**S1 File. GPPS variables extracted and considered in the analysis.**
(DOCX)

**S2 File. Bivariate correlation analysis of patient satisfaction by ethnicity (England: 2019, 2020 & 2021).**
(DOCX)

**S3 File. Stepwise multilevel linear regression results of predictors of patient satisfaction.**
(DOCX)

## Author Contributions

**Conceptualization:** John Paul Magadi, Monica Akinyi Magadi.

**Data curation:** John Paul Magadi.

**Formal analysis:** John Paul Magadi, Monica Akinyi Magadi.

**Investigation:** John Paul Magadi.

**Methodology:** John Paul Magadi, Monica Akinyi Magadi.

**Validation:** Monica Akinyi Magadi.

**Writing – original draft:** John Paul Magadi.

**Writing – review & editing:** Monica Akinyi Magadi.

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
