## [Decision Letter · Decision Letter 0]

30 Sep 2022

PONE-D-22-17218Ethnic inequalities in patient satisfaction with primary health care in the UK: evidence from recent General Practitioner Patient Surveys (GPPS)PLOS ONE

Dear Dr. Magadi,

Thank you for submitting your manuscript to PLOS ONE. After careful consideration, we feel that it has merit but does not fully meet PLOS ONE’s publication criteria as it currently stands. Therefore, we invite you to submit a revised version of the manuscript that addresses the points raised during the review process.

The reviewers have engaged with the study and have recommended a few changes that need to be effected. Please go through the comments, effect the necessary changes and respond to the reviewers as appropriate.

We look forward to receiving your revised manuscript.

Kind regards,

Christmal Dela Christmals, PhD, MSc, BSc, RN

Academic Editor

PLOS ONE

Journal Requirements:

Reviewers' comments:

Reviewer's Responses to Questions

**Comments to the Author**

1. Is the manuscript technically sound, and do the data support the conclusions?

Reviewer #1: Yes

Reviewer #2: Yes

2. Has the statistical analysis been performed appropriately and rigorously? 

Reviewer #1: Yes

Reviewer #2: Yes

3. Have the authors made all data underlying the findings in their manuscript fully available?

Reviewer #1: No

Reviewer #2: Yes

4. Is the manuscript presented in an intelligible fashion and written in standard English?

Reviewer #1: Yes

Reviewer #2: Yes

5. Review Comments to the Author

Reviewer #1: This was a well written manuscript, there were two statements that the researchers made without any reference.

Line 60 - Researchers mentioned interventions that were previously done, it would be interesting to know what these interventions were in order to compare with results of the current study and see if any of those interventions would be applicable.

Line 111-112 the statement made has no reference ; however I saw in the results sections in the researchers did reference teh statement.

Reviewer #2: This study uses publicly available survey data to assess the relationship between the proportion of different ethnic groups and overall satisfaction with primary healthcare within GP practices in England. Other patient-related and service-level factors were included in multilevel models to determine how these impacted on the association between ethnicity and patient satisfaction. While there was a positive association between proportion of survey respondents that were White and patient satisfaction, there was a negative association with other ethnic groups. Adjusting for service-level factors – most notably “care and concern” – attenuated the association between ethnicity and patient satisfaction.

This is a useful piece of work that has updated the research field and provided evidence about the factors which Generally, the analysis is thorough and well-presented. As the surveys are yearly, 3 years is not long enough to examine a trend, although this study provides recent evidence that can be interpreted in the context of previous work. Removing references to the trend element would help to focus on the rest of the analysis.

A key improvement would be to give more detail about the ethnic groups used in the analysis. The surveys included used different ethnic groups, and the list provided in the Annex does not include all of the White ethnic groups. Describing in the Methods or providing a key in the Annex showing which groups were included in the groups analysed would be helpful. Some of the White groups would be described as ethnic minority groups, and have been shown to have different responses to the GPPS than the White British group (e.g. Burt et al Br J Gen Pract 2016;66(642):e47-52) although the numbers would probably have been too small to analyse. Were they included in the broad White group or excluded? Does the “Other Asian” group analysed include the Chinese group or is it just the “Any other Asian background” group? The Asian census category in England and Wales changed to include the Chinese group in 2011, and it is helpful to be clear about the categories used. The terms “Southeast Asians”, “South Asians” and “Asians” are all included in the manuscript – it would be useful to describe the differences between these groups, or use (and define) a single term if they are referring to the same group. I have some other minor comments, described below:

1. The Introduction is very long, and several sections could be omitted as they are addressed in the Discussion. Indeed, the whole manuscript would benefit from some editing to make it easier to read and checking grammatical and spelling errors.

2. There are several statements in the Introduction which need to be supported by references (e.g. lines 56-59).

3. Line 66. It is not obvious how “genetic factors” or “tougher living conditions” could directly influence patient satisfaction. If the implication is that these factors result in poorer health, which is associated with lower satisfaction scores, this should be made clearer.

4. There should be a balance in the language to try and make statements more neutral when there is no evidence of the causes or intention behind study results. For example, lines 111-112 “Therefore, it is plausible that ethnic minorities dealing with a variety of health issues may exhibit their frustrations subconsciously on their patient satisfaction scores […]” could easily be framed as “[…] with a variety of health issues may not be having their complex needs met and report this in their patient satisfaction scores […]”.

5. England and the UK are often used interchangeably – all of the data analysed are from England, and national representation may not mean it is possible to generalise to the UK (line 218).

6. Methods. How were GP practices that were in more than one survey treated? What was the overlap?

7. Line 225. Remove reference to “most recent” as the 2022 survey is now available.

8. The Data Limitations section should be moved to the Discussion, possibly as part of a Strengths and Limitations section.

9. Table 2. This should either be labelled as “Proportion” or the numbers changed to percentages. Please update the ethnicity categories labels to remove “_”. It is also unclear what the N column refers to, please add a description.

10. All Tables and Figures should include the geography and years studied in the title.

11. The (Field 2009) references should be cited fully.

12. I’d suggest removing Table 3 and reporting the mean and 95% confidence intervals in the text.

13. Line 378. Could either report that (r<|0.4|) or (-0.4<r<0.4) and="" both="" clear="" correlations="" it="" make="" moderate.="" negative="" positive="" that="" to="" were="">14. Line 379. Suggest rephrasing “most negative correlation” to “strongest negative correlation”.

15. Table 4. Suggest moving the table to the Appendix. Please update the ethnicity categories labels to remove “_” and describe “minority” better. This could also be simplified, as all the p-values are <0.001 (not 0.000) and the ** label is not needed (or defined).

16. Figure 1. Please add a y-axis title and change the x-axis title to clarify that it’s showing the % of ethnic minority respondents in each GP practice.

17. Figures 2 and 3. Please give the x and y-axes more descriptive titles. It might be easier to read if the scales were converted to percentages.

18. Lines 562 and 646. The ‘Care and concern’ does not necessarily act as a proxy for “cultural differences and communication barriers”. Is there any evidence that these factors influence respondents’ ratings for the ‘care and concern’ question? Similarly (line 635), Ease of using GP website *may* relate to issues of access and language barriers.

19. There should be some discussion in the (Strengths and) Limitations about the difference between individual level ethnicity and proportion of respondents from ethnic groups.</r<0.4)>

6. PLOS authors have the option to publish the peer review history of their article (what does this mean?). If published, this will include your full peer review and any attached files.

Reviewer #1: **Yes: **Dr Nicholin Scheepers

Reviewer #2: **Yes: **Ruth Jack

---

## [Author Response · Author response to Decision Letter 0]

20 Oct 2022

All comments raised by the reviewers have been addressed in the revised manuscript, as clarified in the 'Response to Reviewers' document

---

## [Decision Letter · Decision Letter 1]

5 Dec 2022

Ethnic inequalities in patient satisfaction with primary health care in England: evidence from recent General Practitioner Patient Surveys (GPPS)

PONE-D-22-17218R1

Dear Dr. Magadi,

We’re pleased to inform you that your manuscript has been judged scientifically suitable for publication and will be formally accepted for publication once it meets all outstanding technical requirements.

Kind regards,

Jim P Stimpson, PhD

Academic Editor

PLOS ONE

Additional Editor Comments (optional):

Reviewers' comments:

Reviewer's Responses to Questions

**Comments to the Author**

1. If the authors have adequately addressed your comments raised in a previous round of review and you feel that this manuscript is now acceptable for publication, you may indicate that here to bypass the “Comments to the Author” section, enter your conflict of interest statement in the “Confidential to Editor” section, and submit your "Accept" recommendation.

Reviewer #2: All comments have been addressed

2. Is the manuscript technically sound, and do the data support the conclusions?

Reviewer #2: (No Response)

3. Has the statistical analysis been performed appropriately and rigorously? 

Reviewer #2: (No Response)

4. Have the authors made all data underlying the findings in their manuscript fully available?

Reviewer #2: (No Response)

5. Is the manuscript presented in an intelligible fashion and written in standard English?

Reviewer #2: (No Response)

6. Review Comments to the Author

Reviewer #2: (No Response)

7. PLOS authors have the option to publish the peer review history of their article (what does this mean?). If published, this will include your full peer review and any attached files.

Reviewer #2: **Yes: **Dr Ruth H Jack

---

## [Editor Report · Acceptance letter]

12 Dec 2022

PONE-D-22-17218R1 

Ethnic inequalities in patient satisfaction with primary health care in England: evidence from recent General Practitioner Patient Surveys (GPPS). 

Dear Dr. Magadi:

I'm pleased to inform you that your manuscript has been deemed suitable for publication in PLOS ONE. Congratulations! Your manuscript is now with our production department. 

Kind regards, 

on behalf of

Prof Jim P Stimpson 

Academic Editor

PLOS ONE